# Antimicrobial Resistance among Beta-Hemolytic *Streptococcus* in Brazil: An Overview

**DOI:** 10.3390/antibiotics10080973

**Published:** 2021-08-12

**Authors:** Rosana Rocha Barros

**Affiliations:** Departamento de Microbiologia e Parasitologia, Instituto Biomédico, Universidade Federal Fluminense, Niterói 24210-130, Brazil; rrbarros@id.uff.br

**Keywords:** *Streptococcus agalactiae*, *Streptococcus pyogenes*, *Streptococcus dysgalactiae* subsp. *equisimilis* (*SDSE*), beta-hemolytic streptococci, antimicrobial resistance, Brazil

## Abstract

*Streptococcus pyogenes*, *Streptococcus agalactiae* and *Streptococcus dysgalactiae* subsp. *equisimilis* (*SDSE*) are the beta-hemolytic streptococci species with the most clinical relevance to humans. These species are responsible for several infections, ranging from mild to life-threatening diseases. Although resistance to recommended drugs has not been so critical as detected in other species, it has occurred in diverse regions. In Brazil, it is possible to observe an increasing macrolide and lincosamide resistance trend due to the spread of polyclonal strains. Macrolide–lincosamide–streptogramin B (MLS) resistance phenotypes have been prevalent among *S.* *agalactiae* and *S.* *pyogenes*, while M phenotype (resistance only to macrolides) has prevailed among *SDSE* resistant isolates. Fluoroquinolone resistance is rare in this country, reported only in *S.*
*agalactiae* and *S.*
*pyogenes*. This is due to nucleotide substitutions in *gyrA* and *parC* genes. Reduced penicillin susceptibility and vancomycin resistance, detected in other regions, have not yet been reported in Brazil. Tetracycline is not a therapeutical option, and resistance has occurred at high levels, especially among *S.*
*agalactiae*. These findings highlight the need for continuous monitoring in order to track the occurrence of antimicrobial resistance among beta-hemolytic streptococci species circulating in this country.

## 1. Introduction

Beta-hemolytic streptococci species are important human pathogens, associated with diseases that occur in subjects with different age ranges or health status. Infections are mainly caused by *Streptococcus pyogenes* (Group A streptococci), *Streptococcus agalactiae* (Group B streptococci), and less frequently, *Streptococcus dysgalactiae* subsp. *equisimilis*, *SDSE* (Groups C and G streptococci). *Streptococcus* spp. are members of the human body microbiota, and pathogenic species can colonize asymptomatically, as *S. pyogenes* in the oropharynx or *S. agalactiae* in the genital tract of pregnant women. Infections range from being superficial and mild, such as sore throat and impetigo, to being invasive and life-threatening, such as streptococcal toxic shock syndrome and neonatal meningitis. Immunological complications, rheumatic fever, and acute glomerulonephritis, developed after a streptococcal infection, corroborate the pathogenic burden of such species [1].

Antimicrobial agents are used in virtually all infections due to beta-hemolytic streptococci once diagnosed, regardless of severity, since it reduces symptoms and prevents immunological complications associated with *S. pyogenes* and *SDSE* pharyngitis or impetigo. Penicillin remains the first option for therapy of streptococcal pharyngitis and prophylaxis of *S. agalactiae* neonatal infections. Recommended alternatives include cephalosporins, lincosamides, macrolides, and vancomycin [2,3]. Fluoroquinolones are also an alternative therapy, especially for invasive streptococcal infections [4].

Beta-hemolytic streptococci resistance to recommended antimicrobial agents has been reported in recent decades. While resistance to drugs such as macrolides and lincosamides has been detected in several regions, occurring at different rates, resistance to other antimicrobials, such as beta-lactams, has been rarely or recently reported.

Although uniformly susceptible to penicillin for years, beta-hemolytic streptococci have recently developed resistance to such a drug. Reduced penicillin susceptibility, leading to minimal inhibitory concentration (MIC) of 0.25–1 µg/mL (susceptibility corresponds to MIC of 0.12 µg/mL or lower) was first described in *S. agalactiae* clinical isolates recovered in 1995–2005 in Japan. DNA sequencing made it possible to observe a few nucleotide substitutions in several penicillin-binding protein (PBP)-encoding genes (*pbp* genes). However, only mutations leading to amino acid substitutions in the transpeptidase active site of PBP2x (Q557E and V405A) were associated with reduced susceptibility [5]. Later, *SDSE* isolates with penicillin MIC of 0.5 µg/mL were reported and related to amino acid substitutions in several PBPs. Once again, alterations in the transpeptidase active site of PBP2x (T341P and Q555E) were linked to penicillin reduced susceptibility [6]. Two clinical isolates of *S. pyogenes*, with higher ampicillin and amoxicillin MICs, were recently reported. Unlike what is observed in the other species, nucleotide substitutions were found only in the *pbp2x* gene, leading to T553K in PBP2x [7].

Macrolide resistance among beta-hemolytic streptococci was first reported in 1959 in *S. pyogenes*, and since then, two major resistance mechanisms have been described in the three species addressed here. Target modification by ribosomal methylases encoded by *ermA* and *ermB* genes confers the macrolide–lincosamide–streptogramin B (MLS) resistance phenotype. The MLS phenotype can be constitutive, when the enzyme is continuously synthesized (cMLS), or inducible, when enzyme synthesis is stimulated by a macrolide (iMLS). The expression of efflux pumps, encoded by the *mefA* gene, is related to the M phenotype and confers resistance only to macrolides of 14 and 15 carbon atoms. Macrolide resistance among beta-hemolytic streptococci has been reported in different geographical regions at varied rates [1]. Besides being associated with target modification (MLS phenotypes), resistance to lincosamides may also occur among macrolide susceptible isolates, named the L phenotype. Such a phenotype results from drug inactivation, encoded by the *lnuB (linB)* gene. Another mechanism, characterized by the efflux of drugs, is encoded by the *lsa* gene family. This genotype determines resistance to lincosamide, streptogramin A and pleuromutilins. Resistance only to lincosamide is much less frequent than macrolide resistance, and it has been reported among *S. agalactiae* and *S. pyogenes* isolates [1,8]

Fluoroquinolone resistance among beta-hemolytic streptococci was firstly described in the late 1990s and early 2000s. It is due to nucleotide substitutions in quinolone resistance determinant regions (QRDR) of Gyrase and Topoisomerase IV encoding genes, especially *gyrA* and *parC* [9,10,11]. Among *S. agalactiae*, the first described amino acid changes that led to resistance to fluoroquinolones were S81L and S79F, due to nucleotide substitutions in *gyrA* and *parC*, respectively [9]. Among *S. pyogenes*, amino acid substitutions in ParC (S79A, F, or Y) were associated with fluoroquinolone resistance [10]. *SDSE*, as *S. agalactiae*, has shown nucleotide substitutions in *gyrA*, leading to S81F, L, Y, or C. Mutations in *parC* led to the same substitutions observed among *S. pyogenes* and also in D83G, N, or Y [11]. Resistance to these antimicrobials has been described in different regions but remains at low levels.

Vancomycin resistance among beta-hemolytic streptococci was reported in two epidemiologically non-related clinical isolates of *S. agalactiae*, recovered in different regions of the USA. Isolates were capsular type II and ST 22 by multilocus sequence typing (MLST). Both isolates had vancomycin MIC of 4 µg/mL (susceptibility corresponds to MIC of 1 µg/mL or lower) and harbored a sequence that shared identity with the *Enterococcus faecalis vanG* element [12]. Table 1 summarizes the most relevant resistance mechanisms to recommended antimicrobials.

Brazil is the largest country in Latin America, with a projected 213.2 million inhabitants [13]. Social inequalities are historical and strongly impact population access to health assistance and education. The country is divided into five major geographical regions, South and Southeast being the most developed. These two regions have better human development indexes, such as lower infant mortality rates and longer life expectancy, compared to national rates (11.9 per live 1000 births and 76.6 years, respectively) [14]. While some health threats have a national compulsory notification [15], which contributes to their continuous epidemiological surveillance, others, including several infectious diseases, do not have a national notification system. Regional differences can be observed in data production and analysis, with the majority of studies being conducted in the most developed regions cited above. Regarding therapeutical regimens, it is noteworthy that the Brazilian Health Regulatory Agency (Anvisa) has recently made available national data about antibiotic consumption [16], which may be an important tool to evaluate antimicrobial resistance rates among bacterial pathogens circulating in the country.

Infections due to beta-hemolytic streptococci are not nationally reported, and GBS screening among pregnant women at low risk is not recommended by public health authorities in Brazil [17]. Epidemiological data about infections and antimicrobial resistance rates have been generated by researchers mainly affiliated with academic institutions. In this review, I evaluated available data about antimicrobial resistance rates of beta-hemolytic streptococci species, associated with colonization and infections, that have occurred exclusively in the human population living in this country.

## 2. Results

Twenty-seven published articles about antimicrobial resistance rates of *S. agalactiae* (17), *S. pyogenes* (7), *SDSE* (2), and *S. pyogenes*/*SDSE* (1) were selected, comprising a period of 17 years (2003–2020). All studies followed Clinical and Laboratory Standards Institute (CLSI) guidelines to perform antimicrobial susceptibility testing (AST) and investigate macrolide resistance phenotypes. The detailed steps for article eligibility are shown in Table 2.

### 2.1. Streptococcus agalactiae

*S. agalactiae* represents the beta-hemolytic streptococci with a greater amount of AST data in Brazil. This may reflect its clinical relevance regarding neonatal infections, although national health authorities have not recommended universal screening. Studies were published in 2003–2020, comprising isolates recovered from 1980 to 2018. Most studies were performed in Rio de Janeiro, a Southeast state, followed by Southern states (Paraná, Santa Catarina, and Rio Grande do Sul). Data from all other regions were also available. The majority of *S. agalactiae* isolates were recovered from pregnant women, but isolates from neonates and non-pregnant adults were also included. Resistance to beta-lactams and vancomycin has not been detected. However, resistant isolates to lincosamides, macrolides, and fluoroquinolones have been reported.

Analysis of clindamycin and erythromycin non-susceptibility rates over the years was performed with data from 15 studies [18,19,20,21,22,23,24,25,26,27,28,29,30,31,32], divided according to the period (until and after 2010) and locality of isolates recovery (Table 3). Non-susceptibility refers to resistant and intermediate isolates, according to CLSI definitions and breakpoints. Clindamycin non-susceptibility rates have varied from 1.9 to 18.8%, while erythromycin rates have ranged from 4 to 25%. No statistical difference was observed among resistance rates according to the locality. However, isolates recovered after 2010 were significantly more resistant to both antimicrobials, with mean rates of clindamycin and erythromycin resistance of 6.5% and 6.8%, respectively, until 2010, and 11.3% and 16.2% after 2010. MLS phenotypes, as well as *erm* genes, were prevalent. Two studies [21,31] reported the occurrence of the L phenotype, but no further genetic characterization of such isolates was available since *lnuB* gene investigation failed [21]. Capsular type V has predominated among macrolide-resistant isolates, followed by types Ia and III.

Fluoroquinolone resistance has been a rare event, reported in only three studies since 2011. Resistance rates have varied from 1–7.1% [20,30,31]. Further comprehensive characterization of the first resistant isolates demonstrated a non-clonal relationship among isolates. Point mutations were detected in QRDR of *gyrA* and *parC* genes, leading to S81L in GyrA and S79F in ParC, both associated with levofloxacin resistance [33].

Although tetracycline has not been recommended for the therapy of any infection due to beta-hemolytic streptococci, testing of such a drug was performed in several studies in order to monitor resistance. Tetracycline resistance rates have varied from 75–100%, as observed in 11 studies [18,19,20,21,22,26,27,28,29,30,34]. The mean rate was 87.3%, without any significant differences regarding the period or locality of isolates’ recovery. Genetic determinants of tetracycline resistance were investigated in two studies, being *tetM* the most frequent, followed by *tetO* and *tetL* [19,26]. These genotypes are associated with modified target sites (*tetM* and *tetO*) and efflux pump expression (*tetL*).

### 2.2. Streptococcus pyogenes

Although several papers about the epidemiology of *S. pyogenes* infections have been published, only seven studies that reported AST in isolates circulating in Brazil were eligible for this review. As observed in *S. agalactiae*, most studies were conducted in Rio de Janeiro. Data from São Paulo (Southeast), the Federal District (Central-West region), and Southern states were also available. Studies were published in 2003–2019 and comprised isolates recovered over 24 years (1993–2017). Regarding clinical origin, most isolates were obtained from oropharynx secretion, followed by skin and sterile site specimens.

Reduced penicillin susceptibility has not been reported among *S. pyogenes* isolates. On the other hand, clindamycin and erythromycin non-susceptibility rates have varied from 0.8 to 15.4% (mean 5.5.%) and from 1.6 to 15.4% (mean 5.9%), respectively. Although the M phenotype and *mefA* gene have been predominant in earlier studies, the MLS phenotypes, associated with *erm* genes, became prevalent among macrolide-resistant *S. pyogenes* isolates after 2000 [35,36,37,38,39]. Several *emm* types have been detected among resistant isolates, such as *11*, *12*, *22*, *58*, *73*, and *78* [37,38]. No reports of clindamycin resistance associated with the L phenotype have been found.

Non-susceptibility to fluoroquinolones was reported only in one study, published in 2009 [40]. Eight isolates, 6.1% of a sampling of 130 *S. pyogenes* submitted to AST, presented ciprofloxacin MIC of 2 µg/mL (neither CLSI nor EUCAST have defined breakpoints to this antimicrobial). Isolates showed a polyclonal origin, belonging to several *emm* types, such as *59*, *6*, and *74*. Amino acid substitutions (S79A or F) were found in ParC [40].

Tetracycline resistance has varied from 13.6 to 61%, with a mean rate of 31.8% [35,36,37,38,39,41]. The highest rate was obtained in a study from the Federal District, with isolates recovered in 2004 [41]. Over time, a decreasing trend has been observed, with the lowest rates detected in most recent studies [38,39]. Only the *tetM* genetic determinant was found among tetracycline-resistant isolates, as reported by one study [41].

### 2.3. Streptococcus dysgalactiae subsp. equisimilis

Groups C and G streptococci have been at the center of taxonomic changes in recent decades, being groups C and G with large and beta-hemolytic colonies, recovered from humans, designated as *Streptococcus dysgalactiae* subsp. *equisimilis* [42]. *SDSE* has a high genetic relatedness with *S. pyogenes*, sharing several virulence factors and causing similar clinical manifestations. In this review, only three studies were eligible, being published in 2015–2019. Most of the bacterial isolates were recovered from oropharynx secretion, from 1979 to 2017, in Rio de Janeiro and São Paulo.

As observed in the other species, reduced penicillin susceptibility has not been detected among *SDSE* isolates circulating in Brazil. Clindamycin and erythromycin non-susceptibility rates have varied from 6.8 to 23.1% (mean 13.4%) and from 13.9 to 30.8% (mean 21%), respectively. The M phenotype and *mefA* gene were predominant, followed by iMLS phenotype and *ermA* gene [39,43,44]. Several *emm* types have been associated with macrolide resistance, such as *stG840.0*, *stG653.0*, *stC36.0* and *stC1400.0* [39,44].

Fluoroquinolone resistance has not been reported among *SDSE* circulating in Brazil. On the other hand, the tetracycline resistance mean rate was 37.2% [39,43,44]. As observed in *S. pyogenes*, a decreasing trend has been detected over time. Rates varied from 65.2% in a study with isolates recovered from 1979 to 2008 [43] to 7.7% in the latest study, whose isolates were recovered from 2015 to 2017 [39]. Both *tetK* and *tetM* genes were found in tetracycline-resistant isolates [43].

## 3. Discussion

Antimicrobial resistance is a global health concern with a high burden, especially to some human pathogens, such as the ESKAPE group [45]. *Streptococcus pneumoniae* is the species with the most substantial impact among the streptococci due to the relevant prevalence of invasive pneumococcal infections, such as pneumonia and meningitis, and its resistance to beta-lactams, macrolides, fluoroquinolones, and other antimicrobials [46]. Beta-hemolytic streptococci encompass several human pathogens associated with morbidity and mortality in newborns, children, and adult populations. Although resistance rates to recommended drugs are not as high as in other species, it should also be viewed with concern [46].

As mentioned before, in Brazil, there is neither notification of streptococcal infections nor a recommendation of *S. agalactiae* universal screening among pregnant women [17]. Furthermore, national statistics on antimicrobial resistance rates do not exist. Only very recently, Anvisa has made data available about antimicrobial consumption [16]. However, there is still a lack of information about how frequent antimicrobials have been consumed to treat streptococcal infections in the country.

Reduced penicillin susceptibility has not been reported in Brazil. However, it is essential to highlight that such resistance can only be detected when MIC testing is performed or a specific combination of antimicrobials in disk diffusion testing is used [47].

Resistance to recommended therapeutical alternatives, such as macrolides and lincosamides, has been detected in several studies. Among *S. agalactiae*, increasing resistance, as observed in later studies, is significant. These reports have shown the prevalence of MLS phenotypes and *erm* genotypes, which also confer resistance to clindamycin, one recommended alternative therapy to prevent neonatal infection [3]. Studies performed in other regions have also observed an increasing resistance trend, with the prevalence of MLS phenotypes, as reported in Portugal [48].

Macrolide resistance among *S. pyogenes* isolates has been frequently reported over time, and a change in resistance phenotypes and genotypes prevalence is evident. The M phenotype, predominant in earlier studies, has lately been replaced by MLS phenotypes. As a consequence of such changes, higher rates of clindamycin resistance have been observed in these later studies. MLS phenotypes have also been prevalent in recent studies conducted in other countries, such as in Greece [49]. Among *SDSE* isolates, despite the low amount of data available, it was possible to observe that resistance to macrolides and lincosamides has occurred over time. The M phenotype and *mefA/E* genotype have prevailed among such isolates.

A matter of concern is the recent and widespread usage of azithromycin for “early treatment” of COVID-19 patients in Brazil, as controversially recommended by national health authorities [50]. Considering that streptococci resistance to macrolides has been linked to the consumption of these agents [51], monitoring macrolide resistance in the following years will be critical to verify drug efficacy.

There are few reports of fluoroquinolone resistance among beta-hemolytic streptococci circulating in Brazil, occurring among *S. pyogenes* and *S. agalactiae*. Genetic characterization of resistant isolates, performed in only two studies [33,40], detected nucleotide substitutions in *gyrA* and *parC* genes, as described in studies conducted in other geographical regions.

Tetracycline has been recommended neither to prevent nor to treat infections due to beta-hemolytic streptococci [2,3]. Similarly, CLSI does not include tetracycline in the list of drugs to be tested and reported when AST is performed. However, testing of such antimicrobial has been performed in several studies conducted in Brazil in order to track streptococcal resistance. According to available data, it is possible to observe two patterns over time. While *S. agalactiae* remains broadly resistant, reduced rates have been detected among *S. pyogenes* and *SDSE* isolates in recent studies. Lower resistance rates in these later species may be associated with lesser consumption of tetracycline by Brazilian inhabitants since this antimicrobial is neither a recommended therapy nor can be sold without a prescription, according to a 2011 Anvisa resolution [52]. On the other hand, tetracycline is one of the most relevant antimicrobials used in veterinary settings, where *S. agalactiae* is also an important pathogen, significantly associated with bovine mastitis [53]. Unlike human antibiotic consumption, in Brazil, there is no antibiotic sale control for animal usage.

As one can see, data about antimicrobial resistance in beta-hemolytic streptococci in Brazil are modest. AST’s massive performance must be encouraged in health care settings, followed by notification of unusual resistance phenotypes to reference or research laboratories. This is essential to track antimicrobial resistance and investigate phenotypic and genotypic characteristics of resistant isolates. The generation of national data is vital to improve knowledge about streptococcal infection prevalence and improving therapeutical practices in this population.

## 4. Materials and Methods

Search Terms and Selection Criteria: This review aimed to search and analyze data about antimicrobial resistance rates of beta-hemolytic streptococci associated with colonization and infection in the human population living in Brazil. The analysis included macrolide-resistant phenotypes and genotypes and epidemiological markers of resistant isolates (capsular types of *S. agalactiae* and *emm* types of *S. pyogenes* and *SDSE*). Searching was carried out in PubMed and Lilacs Databases, regardless of the year of publication. Generic search terms were: Streptococcus agalactiae AND antimicrobial resistance AND Brazil; Group B streptococcus AND antimicrobial resistance AND Brazil; Streptococcus pyogenes AND antimicrobial resistance AND Brazil; Group A streptococcus AND antimicrobial resistance AND Brazil; *Streptococcus dysgalactiae* subsp. *equisimilis* AND antimicrobial resistance AND Brazil.

Inclusion criteria: original articles or case reports that provided data about AST, performed according to standard guidelines, of *S. agalactiae*, *S. pyogenes*, and *SDSE* isolates recovered from humans.

Exclusion criteria: studies involving beta-hemolytic streptococci recovered from animals; studies that have not performed phenotypic or genotypic macrolide resistance investigation when erythromycin resistance has been detected. Selection steps are shown in Table 2.

Data of eligible articles were summarized in an Excel sheet. Mean rates were calculated. Statistical analysis, ANOVA (https://goodcalculators.com/one-way-anova-calculator/, accessed on 8 June 2021), and Student *t*-test (https://www.graphpad.com/quickcalcs/ttest1/, accessed on 8 June 2021) were performed to check statistical significance in *S. agalactiae* data.

## 5. Conclusions

This study brings a comprehensive view of antimicrobial resistance among beta-hemolytic streptococci circulating in Brazil. There is no national notification system in this country to monitor streptococcal infection prevalence and antimicrobial resistance rates. However, it is possible to observe that macrolide and lincosamide resistance occurs among all relevant species, which demands a continuous survey. This becomes especially critical with the recent and controversial usage of azithromycin to treat COVID-19. Reliable AST methods must be used in order to detect reduced penicillin susceptibility, the first-line regimen for therapy and prevention of streptococcal infections, which may demand training of personnel and additional costs. In addition, fluoroquinolone and vancomycin resistance surveillance is also important since both drugs are also reliable options for streptococcal infections. Tetracycline resistance occurs at high levels in *S. agalactiae* while having decreased *S. pyogenes* and *SDSE*. A better communication among clinical and reference laboratories is also imperative in order to generate robust data about antimicrobial resistance and, therefore, to guide therapeutical strategies in the country.

## Figures and Tables

**Table 1 antibiotics-10-00973-t001:** Summary of relevant resistance mechanisms to antimicrobial agents recommended to the beta-hemolytic streptococci species addressed in this review.

Antimicrobial Agent	Resistance Mechanism	Genotype	Global Prevalence
Penicillin	Modified target site	Nucleotide substitution in *pbp2x* gene	Reported in all species; rare
Macrolide	Efflux pump	Presence of *mefA* gene	Commonly reported in all species at varied rates
	Modified target site	Presence of *erm* genes	
Lincosamide	Drug inactivation	Presence of *lnuB* gene	Reported in *S. agalactiae* and *S. pyogenes*; rare
	Efflux pump	Presence of *lsa* gene	
Fluoroquinolone	Modified target site	Nucleotide substitution in *gyrA* and *parC* genes	Reported in all species at varied rates
Vancomycin	Modified target site	Presence of *vanG* gene	Reported in *S. agalactiae*; very rare

**Table 2 antibiotics-10-00973-t002:** Published articles in Lilacs and PubMed databases and selection steps.

Articles/Selection Step ^1^	Number
Articles found using search terms	150
Duplicated articles *	72
Abstracts screened	78
Abstracts rejected * (not human subjects; no data about antimicrobial resistance)	39
Articles sought for retrieval	39
Articles not retrieved *	3
Articles assessed for eligibility	36
Articles rejected * (lack of macrolide resistance phenotype or genotype investigation)	9
Eligible articles	27

^1^ Selection steps are highlighted with an asterisk (*).

**Table 3 antibiotics-10-00973-t003:** Prevalence and characteristics of erythromycin and clindamycin non-susceptible *S. agalactiae* recovered in Brazil.

Period of Isolates Recovery	Locality (No. of Studies)	Cli NS%	Ery NS%	Ery Phenotypes ^1^	Ery Genotypes ^2^	Capsular Types ^3^
Until 2010	RJ (5)	4.3–16.7	4.6–13.2	iMLS, cMLS	*ermA*	V, Ia
	BR ^4^ (2)	1.9–3.4	4	cMLS, M	*ermA*, three ^5^	Ia, III
	PR (1)	4.7	4.7	cMLS	*ermA/ermB* ^6^	II, V
After 2010	RJ (3)	3.3–12.2	11.3–14.3	iMLS/cMLS 6, M	*ermA, ermB*	V, III, Ia
	PR (2)	8.1–13.3	8.1–19.3	iMLS	*ermA*	V
	RS (1)	14.2	21.4	M	*mefA*	NT
	BA (1)	18.8	25	cMLS, M	*ermB/mef* ^6^	NT

No., number; Cli, clindamycin; NS, non-susceptible; Ery, erythromycin; RJ, Rio de Janeiro; PR, Paraná; RS, Rio Grande do Sul; BA, Bahia; NT, not tested. ^1^ Prevalent erythromycin-resistant phenotypes in the majority of studies; ^2^ prevalent erythromycin-resistant genotypes in the majority of studies; ^3^ prevalent capsular types in the majority of studies; ^4^ studies conducted in more than one state of Brazil; ^5^ three macrolide resistance genes were simultaneously found (*ermA*, *ermB*, *mefA*); ^6^ simultaneous detection of such phenotype or genotype.

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
