# Peer review of "Antimicrobial Resistance among Beta-Hemolytic Streptococcus in Brazil: An Overview"

_antibiotics, 2021, doi:10.3390/antibiotics10080973_

Round 1

Reviewer 1 Report

While the subject of the article in important and potentially interesting, the data used to draw any conclusions is insufficient. The data were unorganized. There was a table describing increasing resistance only for clindamycin and erythromycin. Data for the remaining drugs was all in text. 

Author Response

To Referee 1. Thank you very much for your revision. Please see in the attachment the answers for your comments.

Reviewer 2 Report

There are some typing in the text. I suggest review. Congratulations on the article.

Author Response

To Referee 2. Thank you very much for your comments.

Reviewer 3 Report

The authors should extend the studies about state of art and background as it is a bit weak and lack to explain the most relevant ones. Besides, a table with some descriptions about materials and methods would be essential for improving it.

Author Response

To Referee 3. Thank you very much for your revision. Please see in the attachment the answers for your comments.

Round 2

Reviewer 1 Report

I have no further issues with the revised manuscript.

Reviewer 3 Report

The article entitled "Antimicrobial Resistance among Beta-Hemolytic Streptococcus in Brazil: an Overview" has a good scientific impact and it is well written, however there is a remark that need to be addressed:

- The Introduction session must be located before materials and Methods.